# Balance Tests in Pre-Adolescent Children: Retest Reliability, Construct Validity, and Relative Ability

**DOI:** 10.3390/ijerph17155474

**Published:** 2020-07-29

**Authors:** Vedrana Sember, Janja Grošelj, Maja Pajek

**Affiliations:** 1Faculty of Sports, University of Ljubljana, 1000 Ljubljana, Slovenia; vedrana.sember@fsp.uni-lj.si; 2Elementary School Spodnja Idrija, 5280 Idrija, Slovenia; janja.groselj@gmail.com

**Keywords:** balance, physical fitness, pre-adolescent children, reliability, validity

## Abstract

Balance is an essential prerequisite for the normal physical development of a child. It consists of the ability to maintain the body’s centre of mass over its base of support, which is enabled by automatic postural adjustments, and maintain posture and stability in various conditions and activities. The present study aimed to determine the measurement characteristics (reliability and concurrent validity) and the relative ability of balance tests and different motor tests in healthy 11-year-olds. We also evaluated the impact of vision on balance ability. Our results showed high interrater reliability (from 0.810 to 0.910) and confirmed the construct validity of the included balance tests. Girls performed significantly better than boys in laboratory tandem stance in following balance components: total sway path with eyes open (BSEO) (*t* = 2.68, *p* = 0.01, effect size (ES) = 0.81), total body sway with eyes closed of centre of pressure (CoP) displacement in the a-p direction (BSEC) (*t* = 1.86, *p* = 0.07, ES = 0.57), mean velocity of CoP displacements (VEO) (*t* = 2.67, *p* = 0.01, ES = 0.83), mean amplitude of CoP displacements in the a-p direction (AapEO) (*t* = 3.38. *p* = 0.00, ES = 1.01) and in mean amplitude of CoP displacements in the m-l direction (AmlEO) (*t* = 3.68, *p* = 0.00, ES = 1.19). With eyes closed, girls performed significantly better (*t* = 2.28, *p* = 0.03, ES = 0.70) than boys did in the mean amplitude of COP displacements in the a-p direction (AapEO) and significantly better (*t* = 2.37, *p* = 0.03, ES = 0.71) in the mean amplitude of COP displacements in the m-l direction (AmlEC). Insignificant correlations between different balance tests, except for a correlation between the flamingo test and one-leg stance on a low beam (*r* = 0.558, p < 0.01), show that each test assesses different aspects of balance ability; therefore, balance cannot be assessed with a single test.

## 1. Introduction

Physical activity (PA) positively affects the overall quality of life [1,2,3,4,5] and is extensively examined to promote healthy lifestyle and sport [6,7]. At the same time, physical fitness (PF), and consequently, fundamental movement skills (FMS) are indispensable part of PA [7]. The development of FMS and subsequent changes in movement proficiency occur in early and middle childhood [8] and are considered to be building blocks that lead to specialised movement sequences required for the adequate participation in various physical activities [9,10]. FMS include locomotor, manipulative or object control, and stability skills [10,11,12]; thus, balance is also an FMS.

Balance is the ability to maintain the body’s centre of mass over its base of support [13] and is enabled by automatic postural adjustments to maintain posture and stability in various conditions and activities [14]. We differentiate between static and dynamic balance. Static balance is defined as the ability to sustain various positions of the contour line and the base of support; its development starts before the 3rd year of age [15]. Balance is dynamic when the person is in motion [16], and its development starts between the 3rd and 7th years. As children age and develop, biomechanical constraints and previous experience change, potentially influencing movement patterns and dynamics. For this reason, balance is considered to be a complex motor skill derived from the interaction of multiple sensorimotor processes [17]. Due to the complexity of human balance, several tests have been developed to assess different aspects of human balance [18,19,20,21,22,23,24]. These tests offer insight into the operation of individual elements of the sensorimotor function in the musculoskeletal system and evaluate the overall physical action quality. Several studies sought links between balance and other FMS in healthy children [25,26,27,28] and researched the balance ability of children with disabilities [29,30,31,32,33,34]. Since we distinguish between two types of equilibrium, we differentiate between tests of static and dynamic balance [35,36,37]. Some research has been conducted on the validity of balance tests, mostly on patients [22,38]; less research is available on the validity of the concepts of static and dynamic balance ability among children or adolescents [32]. 

In contrast, the reliability of balance tests brought researchers [32,39,40,41,42] to conflicting opinions. The flamingo test didn’t show satisfactory reliability results [39,43], whereas walking on a balance beam was selected as a measure of good dynamic balance reliability [32]. Some researchers found possible gender differences in measuring the performance and reliability of individual balance tests; moreover, pre-school girls tend to have better balance abilities [42,43,44,45], while other researchers found no significant links between gender and balance [46,47,48]. When considering which balance testing protocol to apply to assess balance ability, the test should precisely measure the balance task or performance [49]. Moreover, tests should be cheap, easy to use, and performed quickly and practical for clinical or field use. The flamingo balance test is one of the most popular field test to assess static balance [43], since it achieves the requirements of simplicity and low cost, and it is easy to administer in several settings and proper for mass investigations [49].

Taking into consideration the lack of knowledge about validity and conflicting results regarding the reliability of balance field tests on pre-adolescent children, the main aims of this study were as follows:to assess the associations of balance tests with selected motor skills, since the balance is key to all functional movements [28];to assess, select, and include the most appropriate balance tests that measure human balance ability into the PF test battery SLOfit, which for more than 30 years has been an obligatory tool used to evaluate the PF level of school-aged children in Slovenia [50]. The more specific aims of the present study were: (1) to examine possible gender differences in balance; and (2) to distinguish the reliability and validity of non-laboratory balance tests.

## 2. Materials and Methods 

### 2.1. Participants 

The participants were 262 healthy pre-adolescent children in the initial sample (139 boys and 123 girls), aged 11 years ± 6 months, attending Primary School Spodnja Idrija, Idrija, Slovenia (EU). We invited all children aged 11 years (*n* = 268) to participate in our study, within which 6 children were excluded due to medical conditions (*n* = 4) or parental disapproval (*n* = 2). Test–retest reliability was checked in Sample I (*n* = 217; height: 146.4 ± 6.6 cm; weight: 39.1 ± 8.1 kg), where 118 boys (height: 147.2 ± 6.1 cm; weight: 39.1 ± 8.0 kg) and 99 girls (height: 146.7 ± 7.2 cm; weight: 40.1 ± 8.2 kg) participated; and in Sample II (*n* = 45; height: 144.0 ± 7.1 cm; weight: 37.1 ± 8.9 kg), where 21 boys (height: 145.1 ± 8.0 cm; weight: 39.7 ± 9.7 kg) and 24 girls (height: 143.0 ± 6.3 cm; weight: 39.9 ± 7.7 kg) participated. Concurrent validity was checked only on Sample II. Children were randomly assigned to Sample I and Sample II within the Initial Sample (Figure 1). Neurological, locomotor, vestibular, and visual system disorders were used as exclusion criteria. Participants were informed about the purpose of our study, and written informed consent was obtained from all children and their parents. The study was conducted according to the Declaration of Helsinki and approved by the Human Ethics Committee of the Faculty of Sport, University of Ljubljana (363/9.3.2012).

### 2.2. Measurements 

Static balance was measured with the *flamingo test, one-leg stance on a low beam,* and *tandem stance on the force plate*, whereas dynamic balance was measured with the *low-beam walking test*. In the *flamingo test,* the subject stood upright on his or her fully stretched leg on a special wooden beam (50 × 3 × 4 cm), flexed the free leg at the knee, and gripped the foot with the hand on the same side. The timekeeper helped the participant get into the right position and started timing when the subject released the timekeeper’s hand. The result was the maximum number of attempts in 1 min, which was limited to 30. If the subject exceeded this number 15 times in the first 30 s, the subject’s result was 31 [28,51]. In the *one-leg stance on a low beam* test, the subject stepped transversely on a low beam (or reverse Swedish bench) with the dominant foot with hands adducted to the body. The time measurement (altogether 60 s) begun when the subject got into the equilibrium position. The task of the subject was to maintain the equilibrium position as long as possible. The trial was interrupted if the participant touched the beam or stepped with the free leg on the floor. In the best of the equilibrium in tandem stance, displacement of the centre of pressure (CoP) was measured using a force platform (ARS–Analysis and Reporting Software, S2P d.o.o., Ljubljana, Slovenija) with a 1000 Hz sampling rate. In the tandem stance (TS), the dominant (forward) and non-dominant (backward) foot were positioned in a straight line, with toes touching the heel. The result was the movement of a rectangular projection of the centre of gravity of the body expressed in millimetres.

Dynamic balance was measured with the *low-beam walking test*, where the subject stepped into a marked square in front of the beam. On the signal “Go!”, the subject stepped on the low beam (10 cm wide, 390 cm long, and 40 cm high) and walked the length to the other end and back as fast as possible with the hands positioned on the hips. The result was reported in seconds needed to finish the task. All other PF tests, measured with SLOfit test battery, which is the Slovenian national surveillance system for physical and motor development of Slovenian schoolchildren and youth [50,52], are explained elsewhere [28]. All tests, except for the 60-m run and 600-m run, were carried out in a gym. Subjects performed a warm-up prior to testing and performed all tests barefoot on the dominant leg. The order of testing for balance and other PF tests was randomised to minimise the order effect. 

### 2.3. Procedure 

For reliability assessment, the *flamingo test, one-leg stance on a low beam,* and *low-beam walking* tests were performed, since those tests showed the best measurement characteristics in Slovenian settings [28]. Each test was carried out three times, which totaled (3 × 3) nine measurements with breaks of 3–5 min. For concurrent validity, the tandem stance on a force plate was used to measure balance sway and played the role as a gold standard in comparison to the previously mentioned balance tests (Figure 1). The following parameters were calculated: (1) the mean velocity of centre of pressure (CoP); (2) mean amplitudes of CoP displacement in the anterior–posterior and medial–lateral directions; (3) and sway path (the length of the trajectory of CoP displacement in the anterior–posterior and medial–lateral directions) [53]. Furthermore, recurrence quantification analysis (RQA) was performed [54,55]. All the parameters were calculated as average values of the 30 s trial [53].

### 2.4. Statistical Analysis 

Statistical analyses were performed using Microsoft Excell 2016 and Statistical Package for Social Sciences (SPSS, version 22.0 for Windows). For each test, we calculated basic descriptive statistics. Data are presented as mean and standard deviation, *t*-values, Cronbach alpha, correlation coefficients (r), interclass coefficients (ICC) and Cohen’s d as effect size (ES). Data were tested for normal distribution using the nonparametric Shapiro–Wilk’s test (sample II, *n* = 45) and the Kolmogorov–Smirnov test (Sample I, *n* = 217; Sample II, *n* = 262) and were transformed by logarithmic or quadratic transformation, since the data were too deviant from the normal distribution. The association between different balance tests was checked using Spearman’s correlation, whereas correlation between balance tests and other PF tests was checked with Pearson's correlation. Test–retest reliability was determined with interclass correlation coefficient (ICC) and Spearman’s coefficient correlation. The latent structure of the variables was determined by exploratory factor analysis, Bartlett’s test of sphericity, and the Kaiser–Meyer–Olkin criterion. The concurrent validity between laboratory tandem stance and three non-laboratory tests was determined with Spearman’s correlation coefficient. Gender differences and the potential effects of learning or fatigue were calculated using an independent samples *t*-test. 

## 3. Results

The relative ability and descriptive statistics of PF of pre-adolescent children are presented in Table 1. 

We found a weak correlations between coordination (*polygon backwards test*) result and static (*flamingo test* and *one-leg stance on low beam*) and dynamic balance tests (*low-beam walking test*) (from 0.336 to 0.384, *p* < 0.01). The highest—i.e., moderately strong—correlations between coordination indices and balance was found between the *flamingo test* and *polygon backwards* tests in girls (*r* = 0.586, *p* < 0.01). We also noticed weak but significant correlations between explosive strength (*standing broad jump*) and all non-laboratory balance tests (from 0.299 to 0.400, *p* < 0.01), with the highest (a moderately strong) correlation between the *flamingo test* and *standing broad jump tests* in girls (*r* = 0.581; *p* < 0.01). Mostly low correlations were found between the arm and shoulder girdle strength (*bent arm hang*) and all balance tests (from 0.226 to 0.475, *p* < 0.05), with the highest (a moderately strong) correlation between the *one-leg stance on low beam* and *bent arm hang* tests in girls (0.535, *p* < 0.01). The endurance test (running 600 m) correlated with all three balance tests (from 0.320 to 0.411, *p* < 0.01), with strong correlation between the *flamingo* and *600 m running* tests in girls (*r* = 0.662, *p* < 0.01). Sprint (*running 60 m*) correlated with the *flamingo test* in boys (*r* = 0.366, *p* < 0.01) and girls (*r* = 0.531, *p* < 0.01), with the *one-leg stance on a low beam test* in boys (*r* = 0.201, *p* < 0.05) and girls (*r* = 0.337, p < 0.01), and with the *low-beam walking test* in girls (*r* = 0.208, *p* < 0.01) and in boys (*r* = 0.359, *p* < 0.05). All correlations are presented in Table 2.

The results of the Shapiro–Wilk test indicate that the variables *flamingo test*, *one-leg stance on a low beam test*, and *low-beam walking test* do not follow normal distribution (*p* < 0.05). Spearman’s correlation showed a significant moderate correlation only between the *flamingo test* and *one-leg stance on a low beam test* (*r* = 0.558, *p* < 0.01), while the correlation was not significant between laboratory and non-laboratory tests (*p* > 0.05). Spearman’s correlation showed strong or very strong correlation (*r*_*s*1_ = 0.773, *p* = 0.00; *r*_*s*2_ = 0.715, *p* = 0.00; *r*_*s*3_ = 0.810; *p* = 0.00) between attempts in the *flamingo test* (*n* = 217), moderate or strong correlation between attempts in the *one-leg stance on a low beam* (*r_s1_* = 0.598, *p* = 0.00; *r*_*s*2_ = 0.529, *p* = 0.00; *r_s3_* = 0.620; *p* = 0.00), and strong or very strong correlation *(r_s1_* = 0.742, *p* = 0.00; *r_s2_* = 0.699, *p* = 0.00; *r_s3_* = 0.810; *p* = 0.00) between attempts in the *low-beam walking test*. There is a significant correlation between the static *flamingo* and *one-leg stance on a low beam* tests (*r_s_* = −0.559, *p* = 0.00), whereas no significant correlation was found between laboratory and non-laboratory tests and between static and dynamic balance tests. Our results showed the high degree of reliability between three measurements in selected balance tests (*flamingo test*, *one-leg stance on a low beam*, *walking test on a low beam,* and *laboratory tandem stance*) (Table 3). In the *flamingo test*, pre-adolescents made 12.29 ± 6.69 (Sample I) and 12.85 ± 5.83 (Sample II) attempts within a one-minute interval, for which the average ICC values were 0.910 (Sample I) and 0.925 (Sample II) with 95% confidence intervals ranging from 0.888 to 0.929 (Sample I) and 0.877 to 0.956 (Sample II). Pre-adolescents spent on average 30.24 ± 17.35 (Sample I) and 50.42 ± 13.96 (Sample II) seconds in the *one-leg stance on a low beam* position, where the average ICC values in the *one-leg stance on a low beam test* were 0.810 (Sample I) and 0.791 (Sample II) with a 95% confidence interval ranging from 0.888 to 0.929 (Sample I) and from 0.658 to 0.878 (Sample II). In the *low-beam walking test*, pre-adolescents finished on average in 5.9 ± 1.40 (Sample I) and 6.05 ± 1.14 (Sample II) seconds, where the average ICC values in the mentioned dynamic test were 0.882 (Sample I) and 0.925 (Sample II) with 95% confidence intervals ranging from 0.852 to 0.907 (Sample I) and 0.876 to 0.956 (Sample II). All the results, divided by gender, are presented in Table 3. 

Factor analysis revealed the existence of two factors, which explain 62.26% of the total variability in Sample I. The first factor includes all non-laboratory tests that involved maintaining balance on one leg (*one-leg stance* and *flamingo test*) with 20.67% and 41.57% of the total variance explained, respectively. Factor analysis revealed the existence of two factors, which explain 85.03% of the total variability in Sample II. The first factor showed that non-laboratory equilibrium tests explain 31.27% and laboratory variables measured on the force platform explain 53.76% of the total variance. 

A principal components analysis was run on 9 variables that measure balance. The Kaiser–Meyer–Olkin measure of sampling adequacy (χ^2^(36) = 0.845, *p* = 0.00) and Bartlett’s test of sphericity (903.28) (*p* = 0.05) showed that the data are adequate for further validity analysis. The total variance and the scree plot identified two factors explaining 82.81% of the variance and dividing balance ability into the laboratory and non-laboratory components of the measured variables. The first factor explained 67.26% of the total variance and includes all laboratory variables of the tandem stance on the force plate. The strongest variables in this factor are total sway path with eyes open (BSEO), mean velocity of CoP displacements (VEO), and body sway of CoP displacements in the a-p direction (BSapEO). The second factor explains 15.55% of the total variance and includes the non-laboratory balance tests. The strongest variable in the second factor is the flamingo test (FLAM). The values of the the obtained factors are shown in Table 4. 

The independent *T*-test did not reveal differences between boys and girls in non-laboratory balance tests, whereas girls performed significantly better than boys (means and SD in Table 3) in the laboratory tandem stance in the following balance components: BSEO (*t* = 2.68, *p* = 0.01, ES = 0.81), BSEC (*t* = 1.86,*p* = 0.07, ES = 0.57), VEO (*t* = 2.67, *p* = 0.01, ES = 0.83), AapEO (*t* = 3.38, *p* = 0.00, ES = 1.01) and AmlEO (*t* = 3.68, *p* = 0.00, ES = 1.19). With eyes closed, girls (2.89 ± 0.63 mm) performed significantly better (*t* = 2.28, *p* = 0.03, ES = 0.70) than boys (2.49 ± 0.50 mm) in the mean amplitude of COP displacements in the a-p direction (AapEO) and significantly better (*t* = 2.37, *p* = 0.03, ES = 0.71) in the mean amplitude of COP displacements in the m-l direction (AmlEC). 

Independent *T*-tests did not reveal differences between boys and girls in non-laboratory balance tests, whereas girls performed significantly better than boys (means and SD in Table 3) in laboratory tandem stance in the following balance components: BSEO (*t* = 2.68, *p* = 0.01, ES = 0.81), BSEC (*t* = 1.86, *p* = 0.07, ES = 0.57), VEO (*t* = 2.67, *p* = 0.01, ES = 0.83), AapEO (*t* = 3.38. *p* = 0.00, ES = 1.01), and AmlEO (*t* = 3.68, p = 0.00, ES = 1.19). With eyes closed, girls (2.89 + 0.63 mm) performed significantly better (t = 2.28, *p* = 0.03, ES = 0.70) than boys (2.49 ± 0.50 mm) in the mean amplitude of COP displacements in the a-p direction (AapEO) and significantly better (*t* = 2.37, *p* = 0.03, ES = 0.71) in the mean amplitude of COP displacements in the m-l direction (AmlEC). 

## 4. Discussion

To the best of our knowledge, this is the first study to compare retest reliability, concurrent validity, and the relative ability of the *flamingo test*, *one-leg stance on a low beam*, and *low-beam walking test* to the criterion *tandem stance* on the force plate in pre-adolescent children. As far as we know, it comprises one of the largest samples of pre-adolescent children assessed with three non-laboratory and laboratory balance measures, providing test–retest reliability, validity, and relative ability. According to the literature review, most of the research in measurement characteristics has been made on elite athletes in connection with perceptual–motor exercise [56], in the elderly in connection with falls [57], and in children and adolescents with special needs [34,58,59] and disabilities such as blindness, deafness, Down syndrome, polio, ADHD, and injuries [60,61,62,63,64]. However, limited research has been found on healthy, normally developing children and adolescents [65,66]. 

Our results showed that static and dynamic balance tests were associated with explosive strength, power, endurance, and speed, which is in concordance with results elsewhere [28]. Balance performance in girls moderately correlates with the performance in other motor abilities, while in boys, weaker but significant associations could be found. Of all the included balance tests, *the flamingo test* showed the highest associations with the other motor abilities and correlated the best with the endurance running test. Similar results were also found elsewhere in the adult [67] and elderly population [68]. Due to the correlation type of the study, we cannot determine any cause–effect relationship between the balance indices and other motor abilities, but it can be inferred that with improvement in one, we can expect a better result in the other [69]. Therefore, in practical settings, improving these movement foundations would also be associated with improvements of motor skills and possibly balance skills [28]. 

The non-laboratory *flamingo test s*howed a very high reliability score (0.910), whereas the *one-leg stance on a low beam* showed high reliability scores (0.810) in pre-adolescent children. Good measurement properties of the *flamingo test* have also been reported elsewhere [69,70]; the flamingo test was suitable for measuring youth aged 9 to 17 years [71]. We determined that the dynamic *walking test on a low beam* has very high reliability measuring properties (0.910), which is in concordance with the results of other researchers [39] and tends to be a good indicator of dizziness in children and adolescents with neurological problems [72]. The validity of non-laboratory tests showed greater measurement characteristics in laboratory tests compared to non-laboratory tests, which reflects the different approach of obtaining results and the evaluation of the balance tasks between laboratory *tandem stance* and non-laboratory balance tests, for which the criteria most often used are the retention time and the number of attempts to maintain and establish a certain equilibrium position, offering only quantitative results of motor ability and not methods that individuals use to control balance [39,73,74]. Due to the simplicity, low cost, very high reliability, and satisfactory validity scores of the *flamingo test* and *low-beam walking test,* we propose including both tests into battery for the prospective monitoring of child motor development through SLOfit.

Insignificant correlations between different balance tests, with the exception of correlation between the flamingo test and one-leg stance on a low beam (*r* = 0.372, *p* < 0.01), show that each test assesses different aspects of balance ability; therefore, balance cannot be assessed with a single test [32,39]. Based on the obtained results, we propose assessing balance ability with different balance tests simultaneously. Since we differentiate between static and dynamic balance, we recommend the use of one static and one dynamic balance test, such as the *flamingo* and *low-beam walking test*. Both tests are easy to use, cheap, and showed great measurement characteristics in pre-adolescent children.

Girls performed statistically better than boys in the laboratory tandem stance with eyes open and closed. Consistent with the research elsewhere, girls rely less on visual perceptions than boys in this age group do, which may be due to the different physical characteristics and maturity phases at this age. Thus, gender differences can be attributed to the processes of the central nervous system, which do not develop and mature equally rapidly in boys and girls [75]. A recent study [76] also showed significant differences between sexes in posture itself in the prepubertal period because of different body weights and heights, resulting in different lumbar angles and centres of gravity. These findings suggest that girls have better developed balance ability at this age. However, caution should be used, since we are not aware of the cause–effect relationship based on the results of the present study. Nevertheless, the results of the present study support the idea of different teaching strategies for boys and girls during physical education or any other physical activity [77]. 

Our results also showed that the laboratory *tandem stance* was performed better with eyes open than closed in both sexes, showing that visible information is crucial for the successful execution of intentional movements and posture control. People use their vision to predict changes that have an impact on the performance of balance tasks as well to respond to changes that have occurred. With eyes closed, the influx of proprioceptors become the most important for standing regulation [78]. This further suggest that during physical education lessons and any other PA activities, it would be better that children train their balance ability with eyes open and avoid conditions when eyes are closed [77].

### Strengths and Limitations

While these findings contribute significantly to the understanding of postural stability and balance, it should be noted that this study included a sample of healthy pre-adolescents; therefore, results cannot be generalised to samples of varying ages. One major limitation is the small sample size (*n* = 45) in Sample II; therefore, future research should be performed on a larger sample size using laboratory *tandem stance*. Additionally, the different results from different studies could be due to the different testing procedures performed.

Future areas of research should focus on:(i)studying various age groups and patient populations, for instance, a geriatric or population with a neurological handicap, since balance is an essential fundamental movement skill to prevent patients from falling;(ii)exploring the relationship between balance, attention, and mental abilities or cognition at this age group.

## 5. Conclusions

Based on the results of this study, using more than one balance test to assess different aspects of balance is suggested. It is important to take into account that while all of these outcome measures do look at components of balance, none of them can serve as a complete, single evaluative construct of balance itself. Based on the results of the present study, we propose including the *flamingo test* and *low-beam walking test* into the Slovenian physical fitness test battery for the prospective monitoring of child motor development (SLOfit). This study showed that the balance ability is performed better by girls than boys and with eyes open compared to eyes closed, and that visual perceptions in pre-adolescents are very important in controlling their balance. In conclusion, greater emphasis should be placed on the training of endurance, coordination, and muscular strength to improve the balance ability of pre-adolescents. Moreover, training should use different teaching strategies for boys and girls due to gender differences in development and maturity at this age. 

## Figures and Tables

**Figure 1 ijerph-17-05474-f001:**
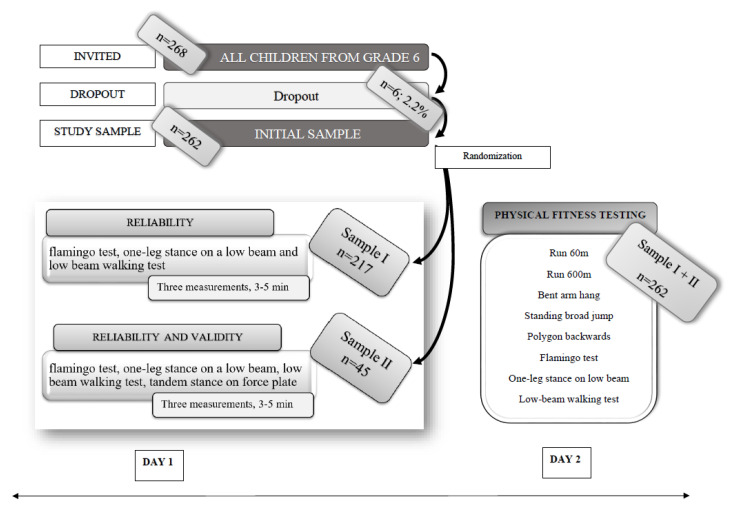
Sampling and measuring procedure flowchart.

**Table 1 ijerph-17-05474-t001:** Physical fitness descriptive statistics for the initial sample.

Gender	*n*	Physical Fitness Test	Mean + (SD)	Physical Fitness Test	Mean ± (SD)
M F ALL	139 123 262	Standing broad jump (cm)	160.04 ± 18.6 156.8 ± 21.5 158.34 ± 20.2	Run 60 m (s)	10.6 ± 8.9 10.9 ± 8.7 10.8 ± 8.9
M F ALL	139 123 262	Bent arm hang (s)	43.4 ± 31.9 39.0 ± 29.4 41.06 ± 30.7	Flamingo test (rep.)	12.6 + 7.0 12.0 + 6.1 12.4 + 6.6
M F ALL	139 123 262	Polygon backwards (s)	13.9 ± 4.0 13.8 ± 3.8 13.8 ± 3.4	One-leg stance on low beam (s)	31.4 + 26.6 35.7 + 36.6 33.7 + 18.4
M F ALL	139 123 262	Run 600 m (s)	162.6 ± 27.3 174.0 ± 28.7 168.7 ± 28.5	Low-beam walking test (s)	5.8 + 1.2 6.0 + 1.5 5.9 + 1.4

**Table 2 ijerph-17-05474-t002:** Correlation coefficients between PF and balance tests for Initial sample.

Balance/PF Test	Gender	*n*	OLSB	LBWT	SBJ	PB	BAH	R60	R600
FLA	M F ALL	139 123 262	−0.376 ** −0.392 ** −0.372 **	0.410 0.363 0.392	−0.304 ** −0.481 ** −.0372 **	0.234 ** 0.586 ** 0.384 **	−0.297 ** −0.429 ** −0.349 **	0.366 ** 0.531 ** 0.430 **	0.320 ** 0.662 ** 0.411 **
OLSB	M F ALL	139 123 262		−0.226 −0.271 −0.238	0.291 ** 0.370 ** 0.299 **	−0.350 ** −0.390 ** −0.362 **	0.447 ** 0.535 ** 0.475 **	−0.201 * −0.337 ** −0.194 **	−0.208 * −0.493 ** −0.411 **
LBWT	M F ALL	118 99 217			−0.360 ** −0.463 ** −0.400 **	0.283 ** 0.427 ** 0.336 **	−0.211 * −0.257 * −0.226 **	0.359 ** 0.208 * 0.301 *	0.268 ** 0.433 * 0.320 **

Legend: PF–physical fitness, FLA–flamingo test, OLS–one-leg stance on a low beam, LBWT–low-beam walking test, SBJ–standing broad jump, PB–polygon backwards, BAH–bent arm hang, R60–running 60 m, R600–running 600 m, *–level of significance 0.01–0.05, **–level of significance < 0.01. Note: all negative results in Table 2 will be presented as positive through the text.

**Table 3 ijerph-17-05474-t003:** Test–retest results for balance and other motor tests.

Balance Tests/Motor Test	Sample	Gender	*n*	Mean + (SD)	ICC	95 % ICC	Cronbach
Flamingo test (rep.)*	I	M F ALL	118 99 217	11.70 ± 6.31 12.99 ± 7.09 12.29 ± 6.69	0.902 0.919 0.910	0.868–0.929 0.887–0.943 0.888–0.929	0.917 0.903 0.910
	II	M F ALL	21 24 45	14.25 ± 6.47 11.63 ± 5.03 12.85 ± 5.83	0.938 0.901 0.925	0.873–0.973 0.806–0.954 0.877–0.956	0.938 0.901 0.925
One-leg stance on low beam (s)	I	M F ALL	118 99 217	28.32 ± 17.63 32.53 ± 16.81 30.24 ± 17.35	0.827 0.785 0.810	0.765–0.856 0.700–0.849 0.762–0.850	0.769 0.838 0.810
	II	M F ALL	21 24 45	46.98 ± 15.28 53.44 ± 12.23 50.42 + 13.96	0.759 0.821 0.791	0.501–0.894 0.646 –0.917 0.658–0.878	0.759 0.821 0.791
Low-beam walking test (s)	I	M F ALL	118 99 217	5.76 ± 1.22 6.81 ± 1.59 5.91 ± 1.41	0.881 0.880 0.882	0.838–0.914 0.832–0.916 0.852–0.907	0.873 0.896 0.882
	II	M F ALL	21 24 45	5.80 ± 1.16 6.27 ± 1.10 6.05 ± 1.14	0.937 0.906 0.925	0.870–0.973 0.815–0.956 0.876–0.956	0.937 0.906 0.925
BSEO (mm)	II	M F ALL	21 24 45	1185.45 ± 71.56 1020.97 ± 26.79 1097.73 ± 59.40	0.897 0.931 0.916	0.787–0.955 0.863–0.968 0.862 –0.951	0.897 0.931 0.916
BSEC (mm)	II	M F ALL	21 24 45	1924.95 ± 456.95 1588.06 ± 388.91 1745.27 ± 450.40	0.780 0.901 0.858	0.545 - 0.904 0.804–0.954 0.767–0.917	0.780 0.901 0.858
BSapEO (mm)	II	M F ALL	21 24 45	881.30 ± 226.91 741.79 ± 201.99 806.89 ± 222.90	0.910 0.941 0.926	0.814–0.961 0.884–0.973 0.878–0.957	0.910 0.941 0.926
BSmlEO (mm)	II	M F ALL	21 24 45	615.68 ± 123.76 558.00 ± 108.09 584.92 ± 117.97	0.843 0.869 0.859	0.675–0.931 0.741–0.939 0.769–0.918	0.843 0.869 0.859
VEO (mm/s)	II	M F ALL	21 24 45	39.51 ± 9.06 34.03 ± 7.56 36.59 ± 8.65	0.896 0.931 0.915	0.786–0.955 0.863–0.968 0.861–0.951	0.896 0.931 0.915
AapEO (mm)	II	M F ALL	21 24 45	5.51 ± 1.98 3.98 ± 1.46 4.69 ± 1.87	0.922 0.920 0.926	0.838–0.966 0.842–0.963 0.879–0.957	0.922 0.920 0.926
AmlEO (mm)	II	M F ALL	21 24 45	4.53 + 1.25 3.39 ± 0.95 3.93 ± 1.23	0.777 0.819 0.827	0.538–0.902 0.642–0.916 0.717–0.899	0.777 0.819 0.827

Legend: * number of repetitions in 60 s, EO–eyes open, EC–eyes closed, BSEO–total body sway with EO of centre of pressure (CoP) displacement in all directions, BSEC–total body sway with EC of CoP displacement in all directions, BSapEO–body sway of CoP displacements in the a-p direction, BSmlEO–body sway of CoP displacements in the m-l direction, VEO–mean velocity of CoP displacements, AapEO–mean amplitude of COP displacements in the a-p direction, AmlEO - mean amplitude of COP displacements in the m-l direction.

**Table 4 ijerph-17-05474-t004:** Component matrix of all nine included components.

	Component	
Balance Variable	1	2
Total sway path with eyes open (BSEO)	0.990	−0.041
Total body sway with EC of CoP displacement in all directions (BSEC)	0.845	−0.097
Body sway of CoP displacements in the a-p direction (BSapEO)	0.972	−0.035
Body sway of CoP displacements in the m-l direction (BSmlEO)	0.891	−0.014
Mean velocity of CoP displacements (VEO)	0.990	−0.040
Mean amplitude of COP displacements in the a-p direction (AapEO)	0.927	0.093
Mean amplitude of COP displacements in the m-l direction (AmlEO)	0.865	0.173
Flamingo test	0.106	0.852
Low-beam walking test	−0.150	0.787

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
