# Peer review of "Balance Tests in Pre-Adolescent Children: Retest Reliability, Construct Validity, and Relative Ability"

_ijerph, 2020, doi:10.3390/ijerph17155474_

Round 1

Reviewer 1 Report

The work is interesting, but some aspects should be taken into account before publication.

Comments and suggestions for Authors:

  1. Materials and Methods:

- Line 67: “The participants were 262 healthy pre-adolescent children (139 boys)”, please add the number of girls participating in the study.

- one of the aim of the study was: “examination of possible gender differences in balance”, in the line 68-69 there is no information about girls.

- line 68-69: please add words height and weight to brackets.

- Why authors include exactly 262 children? How many invitations did you send? Please write more about recruitment of the study group. If it is possible enter recruitment for the study in the Figure.

- There is no information from where the study participants were from. Were these children randomly chosen from schools? from which city / area? Was the study randomized?

  1. Results:

- line 135-148: The authors provide the correlation results in detail, these data should be presented in the table then they will be more readable. Also Please enter the value of the coefficient p when writing about statistically significant values.

  1. Discussion:

Discussion is quite short. Authors should improve the discussion including the latest articles about balance tests.

  1. Keywords: It needs to be sorted in alphabetical order.

Reviewer 2 Report

This manuscript aiming to determine the measurement characteristics (reliability and concurrent validity) and the relative ability of balance tests and different motor tests in healthy eleven-year-olds. While paper is well-written, authors should address the following concerns.

Abstract

  • Explain abbreviature of balance components.

Introduction

  • Line 33. Add this reference, please:

Yanci J, Los Arcos A, Castillo D, Cámara J. Sprinting, Change of Direction Ability and Horizontal Jump Performance in Youth Runners According to Gender. Journal of Human Kinetics. 2017;60(1):199–207.

  • Line 56. Add information to finish this paragraph. What are the practical application of flamingo test for practitioners? Justify its use for assessing the balance…
  • Lines 61-62. I do not agree with this objective: development of appropriate programs and methods to improve balance and proprioception for those children with poor balance skills (balance exercise as prevention and improvement) as for those who are included in high-level sport. I hope to assess the effectiveness of specific balance program in order to examine its effects.

Methods

  • I do not understand why authors perform physical fitness testing(run 60, run 600 m, standing broad jump…) This is important because authors should add another objective. This is a serious flaw of the paper.

Discussion

  • There is a need to add another paragraph related to the physical fitness testing
  • One major concern is to include practical applications in each paragraph of this section.

Conclusion

  • Delete this information: “Balance is a complex construct, and it is recommended that we encourage the utilization of multiple balance assessment tools to capture the entire picture of an individual’s balance”.
  • Add specific practical applications.

Round 2

Reviewer 1 Report

Thank you for the opportunity to review this resubmission.  Authors have done a nice job addressing reviewers' comments. Thank you. I am ok with acceptance.

Reviewer 2 Report

Authors have done a great job. I think the quality of the paper is much better now.